# Thalamic theta phase alignment predicts human memory formation and anterior thalamic cross-frequency coupling

Catherine M Sweeney-Reed[1]*, Tino Zaehle[1], Jürgen Voges[1,2], Friedhelm C Schmitt[1], Lars Buentjen[1], Klaus Kopitzki[1,2], Hermann Hinrichs[1,2,3], Hans-Jochen Heinze[1,2,3], Michael D Rugg[4], Robert T Knight[5,6], Alan Richardson-Klavehn[1]

[1]Department of Neurology and Stereotactic Neurosurgery, Otto von Guericke University, Magdeburg, Germany; [2]Department of Behavioral Neurology, Leibniz Institute for Neurobiology, Otto von Guericke University, Magdeburg, Germany; [3]German Centre for Neurodegenerative Diseases, Otto von Guericke University, Magdeburg, Germany; [4]Center for Vital Longevity and School of Behavioral and Brain Sciences, University of Texas at Dallas, Dallas, United States; [5]Helen Wills Neuroscience Institute, University of California, Berkeley, Berkeley, United States; [6]Department of Psychology, University of California, Berkeley, Berkeley, United States

*For correspondence: catherine.sweeney-reed@med.ovgu.de

**Competing interests:** The authors declare that no competing interests exist.

**Abstract** Previously we reported electrophysiological evidence for a role for the anterior thalamic nucleus (ATN) in human memory formation (*Sweeney-Reed et al., 2014*). Theta-gamma cross-frequency coupling (CFC) predicted successful memory formation, with the involvement of gamma oscillations suggesting memory-relevant local processing in the ATN. The importance of the theta frequency range in memory processing is well-established, and phase alignment of oscillations is considered to be necessary for synaptic plasticity. We hypothesized that theta phase alignment in the ATN would be necessary for memory encoding. Further analysis of the electrophysiological data reveal that phase alignment in the theta rhythm was greater during successful compared with unsuccessful encoding, and that this alignment was correlated with the CFC. These findings support an active processing role for the ATN during memory formation.

## Introduction

Investigations of human memory formation have focused chiefly on the roles of the hippocampus and the neocortex. Animal and human lesion studies suggest that the anterior thalamic nucleus (ATN) may also be involved as a part of an extended hippocampal-cortical encoding system (*Harding et al., 2000*; *Vertes et al., 2001*; *van der Werf et al., 2003*; *Aggleton et al., 2010*; *Aggleton, 2012*). However, the deep location and small size of the ATN has limited direct electrophysiological recording of the ATN during human memory formation (encoding). We had the rare opportunity of recording intracranial electroencephalographic (iEEG) data during memory encoding from human volunteers with intrathalamic electrodes implanted to treat pharmaco-resistant epilepsy (*Sweeney-Reed et al., 2014*). Enhanced frontal cortex-right ATN (RATN) theta phase synchrony and theta-gamma cross-frequency coupling (CFC) in the RATN predicted successful memory encoding. Since the initial work, we have performed additional analyses to delineate the mechanism by which the RATN facilitates memory formation.

Our initial findings of RATN theta-gamma CFC during successful encoding suggested an active role for the RATN in memory formation (*Sweeney-Reed et al., 2014*), because gamma oscillations are believed to reflect local processing (*Jensen et al., 2007*), and the theta rhythm is well-known to be important in memory processing (*Kahana et al., 2001*; *Hasselmo, 2002*; *Klimesch et al., 2004*; *Buzsáki, 2005*; *Fell and Axmacher, 2011*; *Lega et al., 2012*; *Siegle and Wilson, 2014*). We hypothesized that an active role could involve learning-related processes within the RATN, reflected by RATN synaptic modification, and sought an electrophysiological marker of this process. Synaptic modification only occurs when pre- and post-synaptic action potentials occur within a narrow time window (*Paulsen and Sejnowski, 2000*). Phase alignment indicates an orderly temporal relationship between a stimulus and neural activity across successive stimulus epochs. It has been suggested that oscillatory phase alignment—a re-setting of the phase in a particular frequency range within the neuronal population at the recording site with each new stimulus—is required for activity to fit within the narrow temporal window necessary for synaptic plasticity (*Paulsen and Sejnowski, 2000*; *Rizzuto et al., 2003*; *Buzsáki and Draguhn, 2004*; *Düzel et al., 2005*; *Fell and Axmacher, 2011*). More specifically, the long-term potentiation associated with long-term memory encoding has been shown in animal studies to occur only during a particular phase of the theta oscillation (*Fell and Axmacher, 2011*). To address this issue we investigated whether theta phase alignment occurred in the RATN during successful memory formation.

## Results

RATN theta (4–8 Hz) phase alignment was greater during successful than unsuccessful encoding from 0.9–1.1 s poststimulus (two-sided T-test: T = 8.50, p = 0.00015; cluster-size permutation test: p = 0.048) (*Figure 1*, *Figure 1—figure supplement 1*). The difference was seen at an individual level in all 7 participants (*Figure 1*, *Figure 1—figure supplement 2*). Note that our focus was on the RATN, given our previous findings, but we also show phase alignment for other thalamic nuclei.

We calculated the correlation across participants between theta phase alignment at the time when it was greater during successful than during unsuccessful encoding (0.9–1.1 s) and CFC using Pearson's correlation coefficient. We focused on within-RATN theta-gamma coupling, which differed significantly between successful and unsuccessful encoding (*Sweeney-Reed et al., 2014*). Because we previously identified a theta/alpha (7–12 Hz) phase synchrony difference between successful and unsuccessful encoding early in the post-stimulus period, we also calculated the correlation between phase alignment of these oscillations and subsequent CFC. The CFC frequency ranges (means over 7–8 Hz and 40–50 Hz) were selected a priori in light of our previous results (*Sweeney-Reed et al., 2014*). During successful encoding, the correlation between early frontal theta/alpha phase alignment and CFC was significant (r = 0.84, p = 0.019; false discovery rate (FDR) corrected p-value = 0.02 with a maximum FDR level of q = 0.05, to account for the calculation of multiple correlations). No significant correlation was observed during unsuccessful encoding, nor was there a correlation with early RATN or late frontal theta phase alignment during either successful or unsuccessful encoding (*Figure 2B–F*). The correlation was significant, however, between late RATN theta phase alignment and CFC applying the same frequency cut-offs (r = 0.78, p = 0.039). With a maximum FDR level of q = 0.1, the FDR-adjusted p-value was 0.04. These correlations were then directly compared between successful and unsuccessful encoding by calculating a z-score using the Fisher z-transform for each correlation coefficient and comparing the difference, corrected for sample size, to the unit normal distribution; correlation was significantly greater during successful encoding (criterion p < 0.05). Finally, we identified a broadband (4–20 Hz) early poststimulus RATN phase alignment, but unlike later RATN theta phase alignment, neither it nor the event-related potentials (ERPs) differed significantly between successful and unsuccessful encoding (*Figure 3*, *Figure 3—figure supplement 1*). The ERP peak amplitude was correlated with the early RATN phase alignment during successful (r = 0.75, p < 0.0001) and during unsuccessful encoding (r = 0.70, p < 0.0001) (*Figure 3*). There was no significant correlation between frontal ERP amplitude and frontal phase alignment (p > 0.05).

We directed our correlation analysis to investigate an active processing role for the RATN during successful memory formation. Correlations were therefore calculated between measures which reflect local processing and also differed between successful and unsuccessful encoding: RATN theta phase alignment and the previously reported theta-gamma CFC. We have, however, also

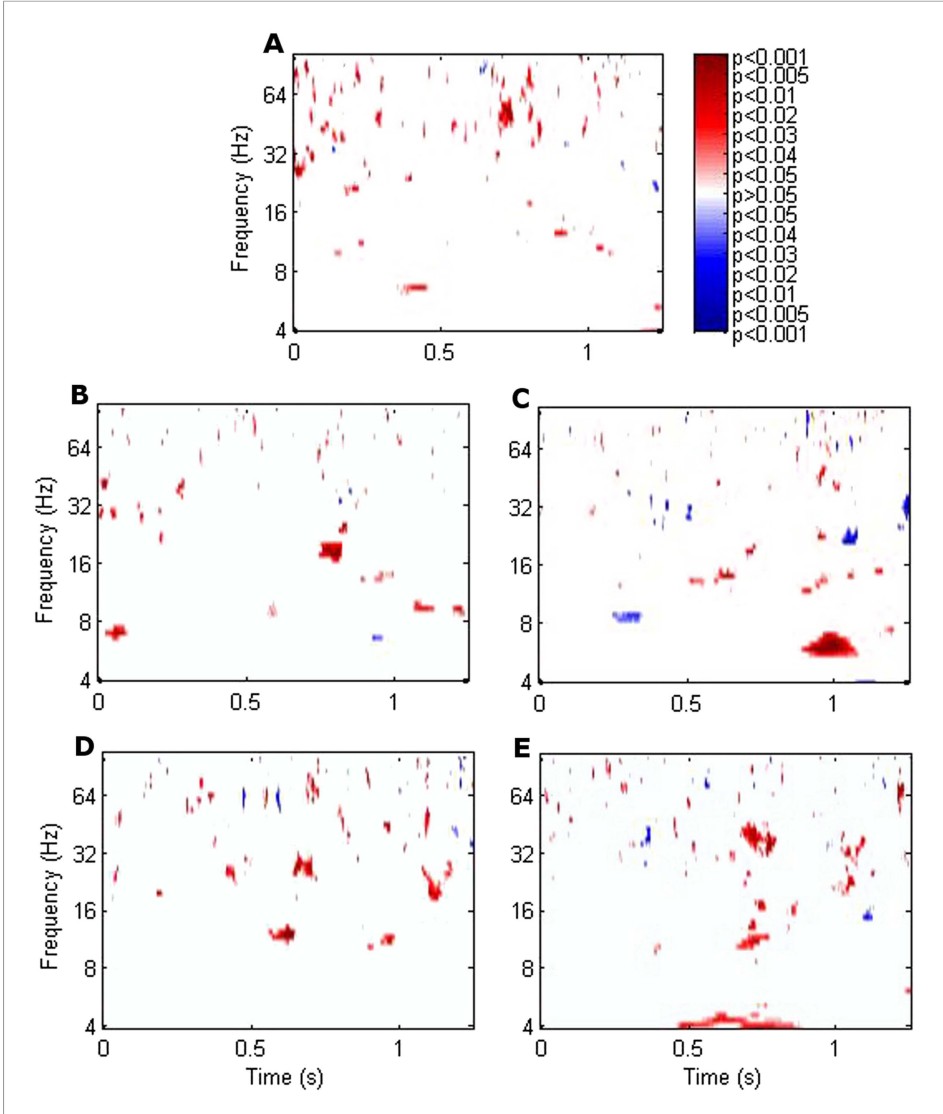

**Figure 1**. Significance of the difference between poststimulus phase alignment during successful compared with unsuccessful encoding using T-tests. (**A**) Frontal. (**B**) Left anterior thalamic nucleus (ATN). (**C**) Right ATN (RATN). At 0.9–1.1 s poststimulus, theta phase alignment was significantly greater during successful compared with unsuccessful encoding (T = 8.50, p = 0.00015). The difference was significant on cluster-size permutation testing (p = 0.048). (**D**) Left dorsomedial thalamic nucleus (DMTN). (**E**) Right DMTN.

The following figure supplements are available for figure 1:

**Figure supplement 1**. Group difference between phase alignment levels during successful vs unsuccessful encoding in the RATN.

**Figure supplement 2**. Mean RATN theta phase alignment 1 s poststimulus was greater following successful than unsuccessful encoding in each individual participant.

analyzed the correlation between phase alignment and the corticothalamic phase synchrony observed previously (*Sweeney-Reed et al., 2014*), although due to the exploratory character of the analyses, we interpret the results with caution. Frontal-RATN theta phase synchrony around 1 s post-stimulus during successful encoding correlated with frontal theta phase alignment (r = 0.88, p = 0.0098) at that time, but not with RATN phase alignment (p > 0.05). Both frontal and RATN

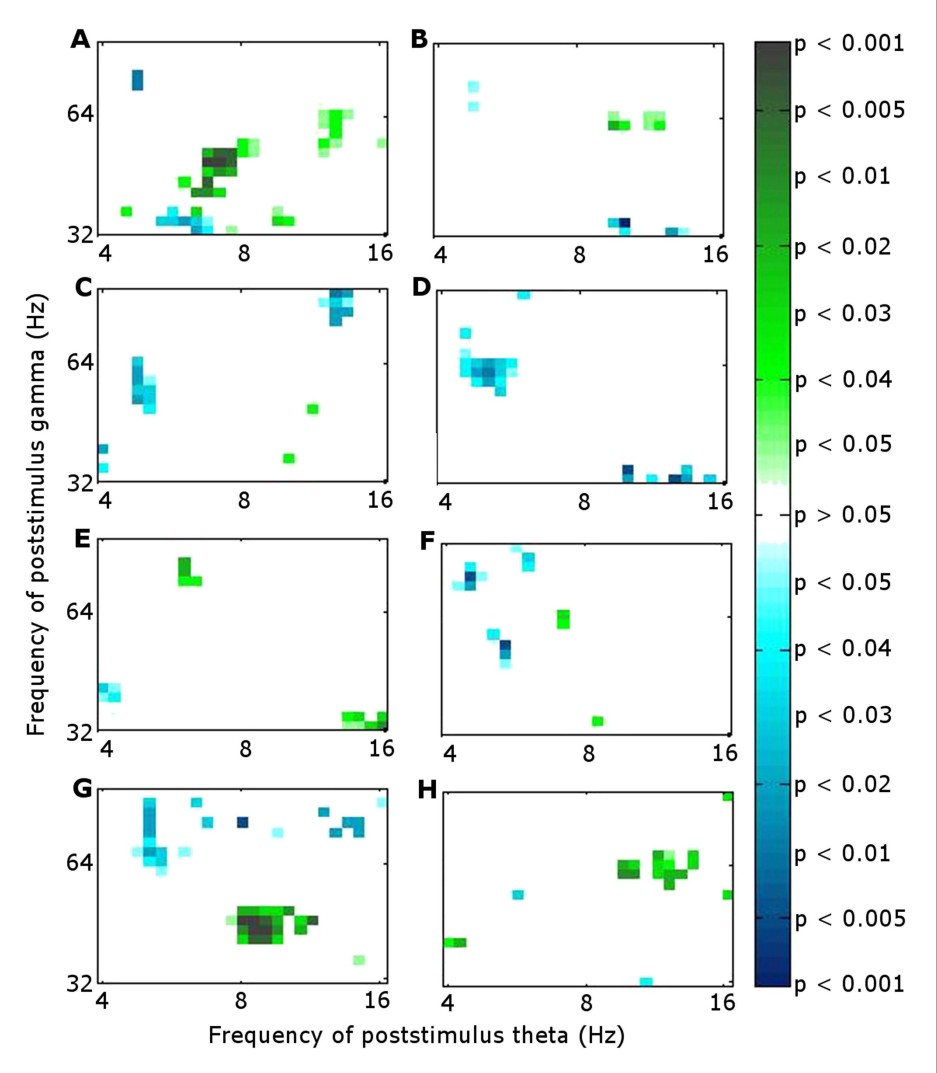

**Figure 2**. Significance of correlation between theta (4–8 Hz) phase alignment and theta-gamma cross-frequency coupling (CFC) within the RATN. (**A**, **B**) Early poststimulus (0–0.5 s) frontal phase alignment: (**A**) Successful encoding (SE): significant correlation (r = 0.84, p = 0.019). (**B**) Unsuccessful encoding (UE): no significant correlation. (**C**, **D**) Early poststimulus RATN phase alignment: (**C**) SE: no significant correlation. (**D**) UE: no significant correlation. (**E**, **F**) Late poststimulus (0.9–1.1 s) frontal phase alignment: (**E**) SE: no significant correlation. (**F**) UE: no significant correlation. (**G**, **H**) Late poststimulus RATN phase alignment: (**G**) SE: significant correlation (r = 0.78, p = 0.039). (**H**) UE: no significant correlation.

theta phase alignment correlated with the early phase synchrony episode across successful and unsuccessful encoding (r = 0.62, p = 0.019 and r = 0.77, p = 0.0012, respectively). Frontal and RATN theta phase alignments were not significantly correlated with each other (p > 0.05) during either the early or the late time periods, consistent with their differing correlations with CFC and phase synchrony.

## Discussion

Our findings provide new information on the role of the RATN in memory encoding. As hypothesized, RATN theta phase alignment was enhanced during successful encoding and was correlated with the theta-gamma CFC we previously identified during memory formation (*Figure 4*).

Human and animal lesion studies indicate that the ATN plays a role in memory encoding, and we provided the direct human electrophysiological support for a key role of the ATN in memory

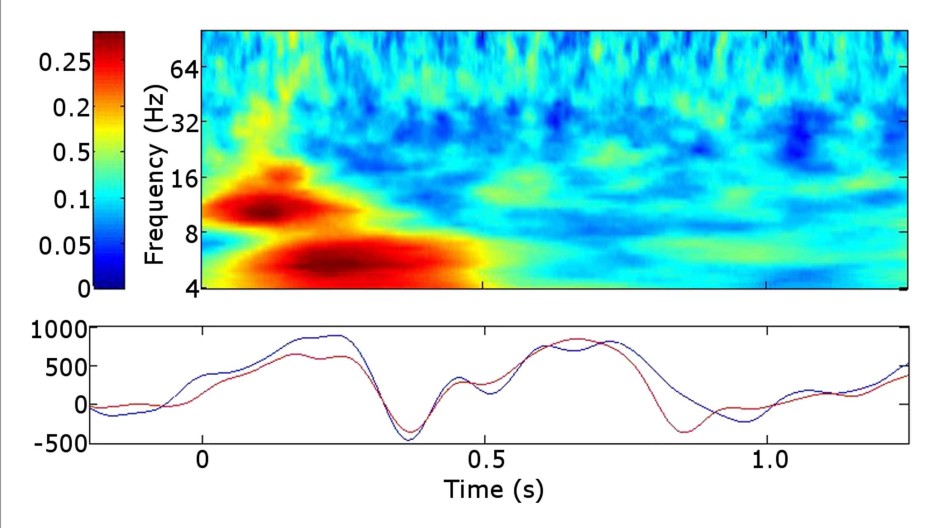

**Figure 3**. Early poststimulus activity in the RATN. Top: Broadband alpha/theta phase alignment early poststimulus during successful encoding (SE). A similar finding during unsuccessful encoding (UE) suggests that this is a task-related phenomenon not specific to successful memory formation. Bottom: Event-related potentials (ERPs). blue = during SE. red = during UE.
The following figure supplement is available for figure 3:

**Figure supplement 1**. Event-related potentials (ERPs).

formation (*Sweeney-Reed et al., 2014*). Here we explore the mechanism by which the RATN modulates memory formation. Theta phase alignment is likely to promote memory-related synaptic plasticity for two reasons. Firstly, phase alignment is deemed a pre-requisite for the synaptic plasticity underlying memory encoding, and secondly, the theta rhythm dominates in the hippocampus, a key structure involved in memory (*Buzsáki, 2005*; *Fell et al., 2011*; *Lega et al., 2012*), with animal studies indicating that hippocampal theta phase influences encoding success (*Hasselmo, 2002*; *Siegle and Wilson, 2014*). We interpret our findings of greater RATN theta phase alignment during successful than unsuccessful memory formation, and its correlation with CFC during successful encoding, as compatible with the suggestion that memory-related changes occur within the RATN itself. The phase alignment in the theta rhythm provides the timing necessary for local synaptic modification and is further reflected in gamma oscillations, thought to reflect local information processing (*Jensen et al., 2007*).

It should be noted that RATN phase alignment could be produced as a downstream consequence of hippocampal phase alignment, and need not derive from independent local RATN processing. Irrespective of its origin, however, the fact that phase alignment occurs in the RATN is suggestive of RATN-specific synaptic modifications. Our interpretation is supported by the finding that integration of head direction and movement information in the ATN in rats suggests an active processing role for the ATN theta rhythm (*Tsanov et al., 2013*), and that hippocampal theta oscillations appear to modulate theta-related plasticity in the ATN (*Tsanov et al., 2011*).

Early post-stimulus frontal theta/alpha phase alignment also predicted subsequent CFC. We interpreted the early phase synchrony as reflecting enhanced item-specific attention and perceptual processing of the stimulus (*Düzel et al., 2005*; *Sweeney-Reed et al., 2014*) and attention allocation has been found to oscillate at a theta rhythm (*Fiebelkorn et al., 2013*). The correlation between the accompanying early frontal theta phase alignment and later CFC fits with the necessity of this early processing for successful memory formation.

The broadband early RATN phase alignment we identified resembled rhinal cortex and hippocampus findings with similar timing reported during both successful retrieval and correct rejection of new items (*Mormann et al., 2005*), interpreted as an unspecific memory processing

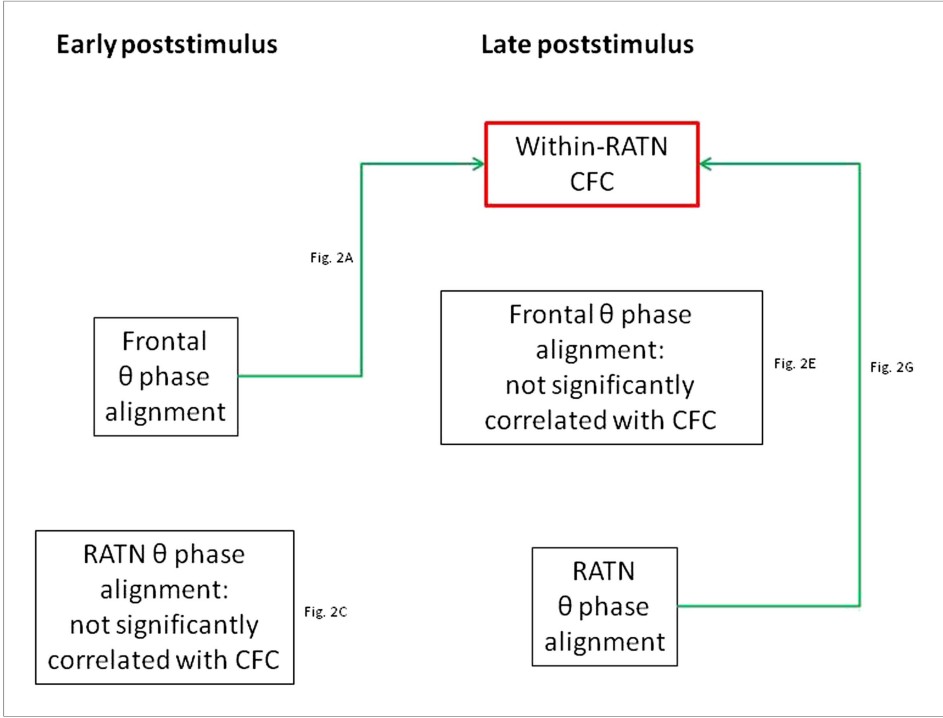

**Figure 4**. Summary of correlations between frontal and RATN theta (θ) phase alignment and within-RATN theta-gamma CFC. Green: significant correlation.

facilitator. The correlation between the early RATN phase alignment and the RATN ERP is consistent with the suggestion that broadband phase re-setting is reflected in the ERP (*Mormann et al., 2005*). Indeed the early RATN phase alignment likely reflects the N1 component of the visual evoked response, which is related to memory performance (*Klimesch et al., 2004*). The similarity between the phase alignment identified here in the RATN and that detected in the rhinal cortex and hippocampus suggests the early phase alignment observed here was also a memory-task-related phenomenon, which is plausible given that a preceding practice session meant our task involved intentional encoding (*Sweeney-Reed et al., 2014*). Our finding provides further support for the hypothesis that the RATN is a part of an extended hippocampal system supporting memory encoding (*Aggleton et al., 2010*; *Sweeney-Reed et al., 2014*).

We tentatively extend our interpretation to the additional correlation analyses involving phase synchrony. Diverse neural correlates of memory formation have been reported around 1 s post-stimulus (*Paller et al., 1987*; *Schott et al., 2002*; *Sederberg et al., 2003*; *Osipova et al., 2006*; *Lega et al., 2012*; *Hanslmayr and Staudigl, 2014*; *Long et al., 2014*), suggesting convergence of different processes underpinning encoding. We postulate that RATN theta oscillations are involved in two separate processes: modulation of frontal-RATN theta oscillations through long-range synchrony and modulation of RATN gamma oscillations through CFC. This suggests an integrative role for the RATN, in accord with the diverse inputs received by this nucleus. Granger causality calculation was consistent with frontal activity driving the theta phase synchrony with RATN activity (*Sweeney-Reed et al., 2014*). We postulate that the correlation between frontal theta phase alignment, which was independent of encoding success, and frontal-RATN theta phase synchrony occurred because frontal theta phase alignment facilitates the phase synchrony. The narrow time window indicated by early frontal theta phase alignment is consistent with an early, rapid perceptual response, and the correlation with the phase synchrony suggests both measures reflect the same process. The correlation between early frontal theta phase alignment and later intra-RATN CFC is consistent with the requirement of early stimulus processing for successful later memory formation processes. We interpret the frontal and RATN phase alignments, which were not correlated with each other, as reflecting different contributions to memory formation. This interpretation is further supported by the

finding that frontal phase alignment was correlated with phase synchrony, whereas the RATN phase alignment was correlated with CFC.

The findings reported here provide further electrophysiological support for the hypothesis that the RATN is actively involved in human memory formation. Alignment of RATN theta oscillatory phase during successful memory encoding provides the temporal precision necessary for synaptic plasticity and correlates with the theta-gamma CFC taking place as a memory is established.

## Materials and methods

Intrathalamic signals from the ATN and dorsomedial thalamic nuclei bilaterally and neocortical activity from simultaneous surface EEG were recorded from 7 human volunteers undergoing electrode placement for subsequent stimulation treatment of pharmacoresistant focal epilepsy. Frontal electrodes, whose placement was limited by surgical dressing location, were located at Fz (frontal), AFz (anterior frontal), or Fpz (frontopolar). The reported data were taken from the most frontal available electrode from each participant as previously reported (*Sweeney-Reed et al., 2014*). The electrophysiological data were recorded during presentation of a series of photographic scenes, which were judged as indoors or outdoors. Each scene was labeled as successfully or unsuccessfully encoded on the basis of a subsequent recognition test, in which the scenes were presented again, mixed with new scenes, and judged as 'old' or 'new'. The thalamic data were re-referenced offline by subtracting the deeper neighbor rendering a bipolar reference. Further details regarding the experimental paradigm, the patients and their diagnoses, electrode placements, data pre-processing, and CFC calculation based on work by *Canolty et al. (2006)*; *Axmacher et al. (2010)* are provided by *Sweeney-Reed et al. (2014)*.

Phase alignment was obtained by calculating the phase-locking factor, which is a stimulus-locked average of phase distribution across trials at a given time-frequency (*Tallon-baudry et al., 1996*). For each time-frequency point, alignment of phase to stimulus onset was calculated across epochs for each participant. The phase alignment index (0 = no phase alignment; 1 = complete phase alignment), was obtained by dividing the length of the unit phase vector across epochs for each participant by the number of epochs (*Düzel et al., 2005*). Numbers of trials were balanced for successful and unsuccessful encoding.

Two-sided T-tests were used to compare phase alignment during successful vs unsuccessful encoding. Due to high dependency between adjacent time-frequency points in the phase alignment analysis, a Bonferroni correction was deemed overly conservative. We therefore examined significance using a cluster-size permutation test, whereby the cluster-sizes of adjacent time-frequency points significant on T-tests were compared with a distribution of cluster-sizes arising by chance (*Maris and Oostenveld, 2007*), analogously to the synchrony analyses (*Sweeney-Reed et al., 2014*). Phase alignment values across successful and unsuccessful encoding were pooled and assigned to two artificial conditions 1000 times, and T-tests were applied each time comparing the two scrambled conditions. Adjacent time-frequency points where $p < 0.05$ were summed providing a value for the largest cluster in each iteration, yielding a two-sided distribution, against which to compare the cluster-size of significant p-values comparing successful with unsuccessful encoding.

In order to minimize the risk of type I errors, we focused our analysis on a hypothesis based on the literature and on our previous findings, which were corrected for multiple comparisons. Because coupling between theta and gamma oscillations differed according to encoding success, we hypothesized that phase alignment, which accompanies synaptic modification, would be correlated with this coupling. Although we could have justified theory-based focus on phase alignment in the theta frequency range, we included frequencies ranging from theta to gamma, performing the cluster-size permutation test described above. We carried out FDR correction to account for the calculation of multiple correlations (*Benjamini and Hochberg, 2009*).

To calculate ERPs, the raw data were filtered from 1–8 Hz, a 200 ms baseline was subtracted from each trial, and an average over trials was calculated for each participant (*Szczepanski et al., 2014*). Correlation was calculated between the ERP and phase alignment during the first 0.5 s poststimulus during successful and during unsuccessful encoding.

# Additional information

## Funding

The authors declare that there was no external funding for this work.

## Author contributions

CMS-R, Conceived and designed the research, Prepared the experiment, Analyzed and interpreted the electrophysiological data, including writing data-analysis programs, Drafted and revised the article; TZ, Acquired the electrophysiological data, Revised the article; JV, Acquired data: selected and provided the technical facilities used for the human stereotactic neurosurgery and intracranial recording, recommended and supervised patient treatment with deep brain stimulation, surgically implanted the intracranial electrodes, Analyzed and interpreted data: localized the intracranial electrodes, Revised the article; FCS, Acquired data: diagnosed the patients, recommended and supervised patient treatment with deep brain stimulation, Revised the article; LB, KK, Analyzed and interpreted data: localized the intracranial electrodes, Revised the article; HH, H-JH, Acquired data: selected and provided the technical facilities used for the human stereotactic neurosurgery and intracranial recording, Analyzed and interpreted data: commented on the data analysis, Revised the article; MDR, RTK, AR-K, Analyzed and interpreted data: advised on the data analysis, Advised on and revised the article

## Author ORCIDs

Catherine M Sweeney-Reed, http://orcid.org/0000-0002-3684-1245

## Ethics

Human subjects: The measurements were approved by the Ethics Commission of the Medical Faculty of the Otto-von-Guericke University, Magdeburg (application number 0308), and all participants gave written informed consent in accordance with the Helsinki Declaration of 1975, as revised in 2000 and 2008. Consent to participate in our study, as well as for publication of results in an anonymized format, was obtained by the neurosurgeon at the same time as consent was obtained for the surgical procedure.

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
