## [Decision Letter]

Thank you for sending your work entitled “Thalamic theta phase alignment predicts human memory formation and anterior thalamic cross-frequency coupling” for consideration at *eLife*. Your article has been favorably evaluated by Eve Marder (Senior editor), a Reviewing editor and three reviewers.

The following individuals responsible for the peer review of your submission have agreed to reveal their identity: Howard Eichenbaum (Reviewing editor); Jurgen Fell, John Aggleton, and Joel Voss (peer reviewers).

The Reviewing editor and the reviewers discussed their comments before we reached this decision, and the Reviewing editor has assembled the following comments to help you prepare a revised submission.

This Research Advance paper builds on a previous study (29) which has demonstrated memory-related theta phase synchronization between frontal and anterior thalamic activity and theta-gamma phase-amplitude coupling within anterior thalamus. Here, increased memory-related inter-trial theta phase alignment of frontal and anterior thalamic activity is demonstrated. Furthermore, positive interindividual correlations between early frontal phase alignment, as well as late anterior thalamic phase-alignment with anterior thalamic theta-gamma phase-amplitude coupling are reported. This article is based on a precious data set, i.e. intracranial thalamic recordings in epilepsy patients performing a subsequent memory paradigm. The analysis techniques are timely and the reported results are important and represent a significant advance and supplement to the previous paper.

However, several substantive concerns were expressed by the reviewers:

1) The rationale underlying the analysis approaches should be explained more clearly:

In the Introduction: “We hypothesized that RATN theta phase alignment may provide the mechanism by which RATN synaptic modification required for memory formation occurs…”. Please explain why RATN synaptic modifications are expected to be necessary for memory formation;

Also in the Introduction: “Secondly, it has been suggested that phase alignment is required for activity to fit within the narrow temporal window necessary for synaptic plasticity” and;

In the Discussion: “Firstly, phase alignment is deemed a prerequisite for the synaptic plasticity underlying memory formation”. Please elaborate on this idea.

2) Much of the study cannot be understood without scrutiny of the previous study, rather begging the question why this had to be a separate paper. Results should be described more thoroughly and should be better integrated with the findings of the previous paper (29):

Please provide ERP curves not only for Figure 3, but also for Figure 1.

Are frontal theta phase alignments and anterior thalamic theta phase alignments interindividually correlated?

Are phase alignments interindividually correlated with the previously reported corticothalamic phase synchrony values (29)?

In the Discussion, the authors state: “The early RATN ERP also did not differ significantly…and was correlated with the early phase alignment…”. Which ERP measure has been extracted, is it peak amplitude? I guess early phase alignment means early RATN phase alignment? This sentence belongs to the Results section.

Are frontal ERP amplitudes correlated with frontal phase alignments?

Figure 4: How does this scheme fit together with the findings, apart from RATN cross-frequency coupling, described in the previous paper (29)?

3) There is concern over the possibility of Type 1 errors. It is clear that the authors have, in places, tried to correct these (as is needed) but quite how they have done this is difficult to follow in places. More clarification would help.

---

## [Author Response]

*1) The rationale underlying the analysis approaches should be explained more clearly*:

*In the Introduction: “We hypothesized that RATN theta phase alignment may provide the mechanism by which RATN synaptic modification required for memory formation occurs…”. Please explain why RATN synaptic modifications are expected to be necessary for memory formation*;

Our hypothesis was that RATN synaptic modification is required for memory formation, and our assumption was that were this the case, we would expect to see local evidence, including increased RATN phase alignment during successful encoding, in addition to the theta-gamma CFC already reported. We thank the reviewers for pointing out the lack of clarity. We have now expanded on this point in combination with our response to the point below.

*Also in the Introduction: “Secondly, it has been suggested that phase alignment is required for activity to fit within the narrow temporal window necessary for synaptic plasticity” and*;

*In the Discussion: “Firstly, phase alignment is deemed a prerequisite for the synaptic plasticity underlying memory formation”. Please elaborate on this idea*.

We have now elaborated on this idea in the second paragraph of the Introduction.

*2) Much of the study cannot be understood without scrutiny of the previous study, rather begging the question why this had to be a separate paper. Results should be described more thoroughly and should be better integrated with the findings of the previous paper (*[29]*)*.

The current manuscript contains further analysis, performed since publication of the original paper. The reviewers are correct that the previous study is necessary for a complete understanding of the current one, which is why we found the Research Advance format offered by *eLife* ideal for communicating the results of further analysis of the data since the original publication. Although the Research Advance would be indexed separately, the two would also be linked, and the instructions state that the Introduction should therefore be minimal. We thank the reviewers for pointing out the difficulty in reading this manuscript in isolation, however, because it should also be able to stand alone, and we have sought to improve the balance with additional information. We now provide more details regarding the electrode locations and the experimental paradigm in the first paragraph of the Materials and methods section. We have also added new paragraphs to the Discussion section, in which we describe the results more thoroughly and integrate the findings of the two papers.

*Please provide ERP curves not only for*
Figure 3*, but also for*
Figure 1.

The ERP curves for Figure 1 are now shown in Figure 1–figure supplement 3.

Are frontal theta phase alignments and anterior thalamic theta phase alignments interindividually correlated?

We have added the following sentence to the Results section:

“Frontal and RATN theta phase alignments were not significantly correlated with each other (p > 0.05) during either the early or the late time periods, consistent with their differing correlations with CFC and phase synchrony.”

We have added the following to the Discussion section:

“We interpret the frontal and RATN phase alignments, which were not correlated with each other, as reflecting different contributions to memory formation. This interpretation is further supported by the finding that frontal phase alignment was correlated with phase synchrony, whereas the RATN phase alignment was correlated with CFC.”

*Are phase alignments interindividually correlated with the previously reported corticothalamic phase synchrony values (*[29]*)?*

We directed our correlation analysis to investigation of an active processing role for the RATN during successful memory formation, calculating correlation between measures likely to reflect local processing, which both differed significantly between successful and unsuccessful encoding, i.e., theta phase alignment in the RATN and the previously detected theta-gamma CFC. The reviewers are right that other correlations within the data are of interest, but we focused our analysis, because calculating all possible correlations within the data would elevate the chance of type I errors. The additional results, however, with our cautious interpretation, are as follows:

“Frontal-RATN theta phase synchrony around 1 s post-stimulus during successful encoding correlated with frontal theta phase alignment […] consistent with their differing correlations with CFC and phase synchrony.”

“We tentatively extend our interpretation to the additional correlation analyses involving phase synchrony. […] This interpretation is further supported by the finding that frontal phase alignment was correlated with phase synchrony, whereas the RATN phase alignment was correlated with CFC.”

*In the Discussion, the authors state: “The early RATN ERP also did not differ significantly…and was correlated with the early phase alignment…”. Which ERP measure has been extracted, is it peak amplitude? I guess early phase alignment means early RATN phase alignment? This sentence belongs to the Results section*.

The reviewers are correct that we extracted peak amplitude and were referring to early RATN phase alignment. We have added these two clarifications and moved the sentence to the Results section.

Are frontal ERP amplitudes correlated with frontal phase alignments?

There was no significant correlation between frontal ERP amplitude and frontal phase alignment (p > 0.05). We have added this finding to the second to last paragraph of the Results section.

Figure 4*: How does this scheme fit together with the findings, apart from RATN cross-frequency coupling, described in the previous paper (*[29]*)?*

We have now combined the current findings with those in the previous paper more extensively, as described above, adding our interpretation to the second to last paragraph of the Discussion section. Because our original analyses were based on our hypothesis, we have retained the summary of these in Figure 4.

*3) There is concern over the possibility of Type 1 errors. It is clear that the authors have, in places, tried to correct these (as is needed) but quite how they have done this is difficult to follow in places. More clarification would help*.

We thank the reviewers for pointing out that more clarity is needed and have elaborated further on these points in the Materials and methods section as follows:

“In order to minimize the risk of type I errors, we focused our analysis on a hypothesis based on the literature and on our previous findings […]. We carried out FDR correction to account for the calculation of multiple correlations (4).”